# Computations underlying *Drosophila* photo-taxis, odor-taxis, and multi-sensory integration

Ruben Gepner[1†], Mirna Mihovilovic Skanata[1†], Natalie M Bernat[1], Margarita Kaplow[2], Marc Gershow[1,2*]

[1]Department of Physics, New York University, New York, United States; [2]Center for Neural Science, New York University, New York, United States

**Abstract** To better understand how organisms make decisions on the basis of temporally varying multi-sensory input, we identified computations made by *Drosophila* larvae responding to visual and optogenetically induced fictive olfactory stimuli. We modeled the larva's navigational decision to initiate turns as the output of a Linear-Nonlinear-Poisson cascade. We used reverse-correlation to fit parameters to this model; the parameterized model predicted larvae's responses to novel stimulus patterns. For multi-modal inputs, we found that larvae linearly combine olfactory and visual signals upstream of the decision to turn. We verified this prediction by measuring larvae's responses to coordinated changes in odor and light. We studied other navigational decisions and found that larvae integrated odor and light according to the same rule in all cases. These results suggest that photo-taxis and odor-taxis are mediated by a shared computational pathway.

**\*For correspondence:** mhg4@nyu.edu

[†]These authors contributed equally to this work

**Competing interests:** The authors declare that no competing interests exist.

## Introduction

Many small organisms navigate their environments by decoding temporal variations in receptor activity to bias motor output decisions (*Berg and Brown, 1972*; *Sawin et al., 1994*; *Pierce-Shimomura et al., 1999*; *Ryu and Samuel, 2002*; *Fishilevich et al., 2005*; *Louis et al., 2008*; *Luo et al., 2008*, *2010*; *Albrecht and Bargmann, 2011*; *Gomez-Marin et al., 2011*; *Lockery, 2011*; *Gershow et al., 2012*; *Kane et al., 2013*). The dynamics of these organisms' decision-making are intimately linked to the properties of the underlying chemical or neural substrates (*Segall et al., 1986*; *Korobkova et al., 2004*; *Miller et al., 2005*; *Chronis et al., 2007*; *Clark et al., 2007*; *Emonet and Cluzel, 2008*; *Suzuki et al., 2008*; *Asahina et al., 2009*; *Shimizu et al., 2010*; *Bretscher et al., 2011*; *Busch et al., 2012*; *Kato et al., 2014*; *Luo et al., 2014*). In their natural environments, animals are confronted with variable and frequently conflicting inputs arriving via multiple sensory pathways. How these simple organisms respond to multi-modal stimuli and how their neural circuits integrate and prioritize multi-sensory information remains unknown.

We sought to decode the computations underlying the *Drosophila* larva's response to visual and olfactory cues, when presented individually or in combination. The second instar larva uses a similar strategy for navigating environments with spatially varying odor concentrations, light levels, and temperatures. Larvae move forward in a series of relatively straight *runs* interspersed with reorienting *turns*, and increase the frequency and magnitude of their turns in response to unfavorable changes in the stimulus (*Sawin et al., 1994*; *Busto et al., 1999*; *Scantlebury et al., 2007*; *Louis et al., 2008*; *Luo et al., 2010*; *Gomez-Marin et al., 2011*; *Gershow et al., 2012*;

**eLife digest** Living organisms can sense cues from their surroundings and respond in appropriate ways. For example, animals will often move towards the smell of food or away from potential threats, such as predators. However, it is not fully understood how an animal's nervous system is set up to allow sensory information to control how the animal navigates its environment. It is also not clear how animals 'decide' what to do when they receive conflicting information from different senses.

Optogenetics is a technique that allows neuroscientists to control the activities of individual nerve cells simply by shining light on to them. Fruit fly larvae have a simple but well-studied nervous system, and they are nearly transparent, so scientists can use optogenetics to activate nerve cells in freely moving larvae.

Fruit fly larvae move in a series of forward 'runs' and direction-changing 'turns' and use sensory cues to decide when to turn, how large of a turn to make, and whether to turn left or right. Gepner, Mihovilovic Skanata et al. used optogenetics to stimulate different combinations of sensory nerve cells in larvae, while tracking the larvae's movements to discover exactly what information they used to make these decisions. An independent study by Hernandez-Nunez et al. also used a similar approach.

Fruit fly larvae are attracted towards scents from rotting fruit and are repelled by light—in particular, larvae are most sensitive to blue light but cannot detect red light. Therefore, Gepner, Mihovilovic Skanata et al. could expose the larvae to blue light to activate light-sensing nerve cells as normal, and use red light to activate odor-sensing nerve cells via optogenetics. These experiments showed that larvae changed direction more often when the level of blue light was increased or when the level of red light (which simulated the detection of odors from rotting fruits) was decreased.

Analysis of the data from these experiments revealed that larvae essentially assign negative values to the blue light and positive values to the 'odor-mimicking' red light. The larvae then use the sum of these two values to dictate their next move. This suggests that navigation in response to both light and odors is supported by the same pathways in a larva's nervous system.

The approach of using optogenetics in combination with quantitative analysis, as used in these two independent studies, is now opening the door to a more complete understanding of the connections between the activities of sensory nerve cells and perception and behavior.

Kane et al., 2013; Klein et al., 2014). During turns, larvae survey the local environment using side-to-side head sweeps to determine the direction of the next run (Sawin et al., 1994; Luo et al., 2010; Gomez-Marin et al., 2011; Gershow et al., 2012; Kane et al., 2013; Klein et al., 2014). That the larva uses the same strategy to respond to a variety of stimuli suggests navigation may be mediated by a single circuit that combines input from the various sensory organs. Alternately, this apparent commonality might result from the larva's limited repertoire of motor outputs with which to implement a navigational response. Perhaps, independent circuits mediate the 'decision' portion of each navigational algorithm and only converge at the motor output level (Frye and Dickinson, 2004).

We sought a computational model that could describe the transformation from sensory input to navigational decisions for uni- and multi-modal stimuli and that could differentiate between shared and parallel navigational circuits. The Linear-Nonlinear-Poisson (LNP) model (Chichilnisky, 2001; Dayan, 2001; Ringach and Shapley, 2004; Bialek and van Steveninck, 2005; Schwartz et al., 2006; Kim et al., 2011) is widely used to relate time-varying input to stochastic output. In this model, decisions (e.g., to initiate a turn, Figure 1A,B) are generated according to a Poisson process whose underlying rate at time $t$ is:

$$r(t) = f[(A*S)(t)]; \; (A*S)(t) = \int_0^\infty A(\tau)S(t-\tau)d\tau,$$

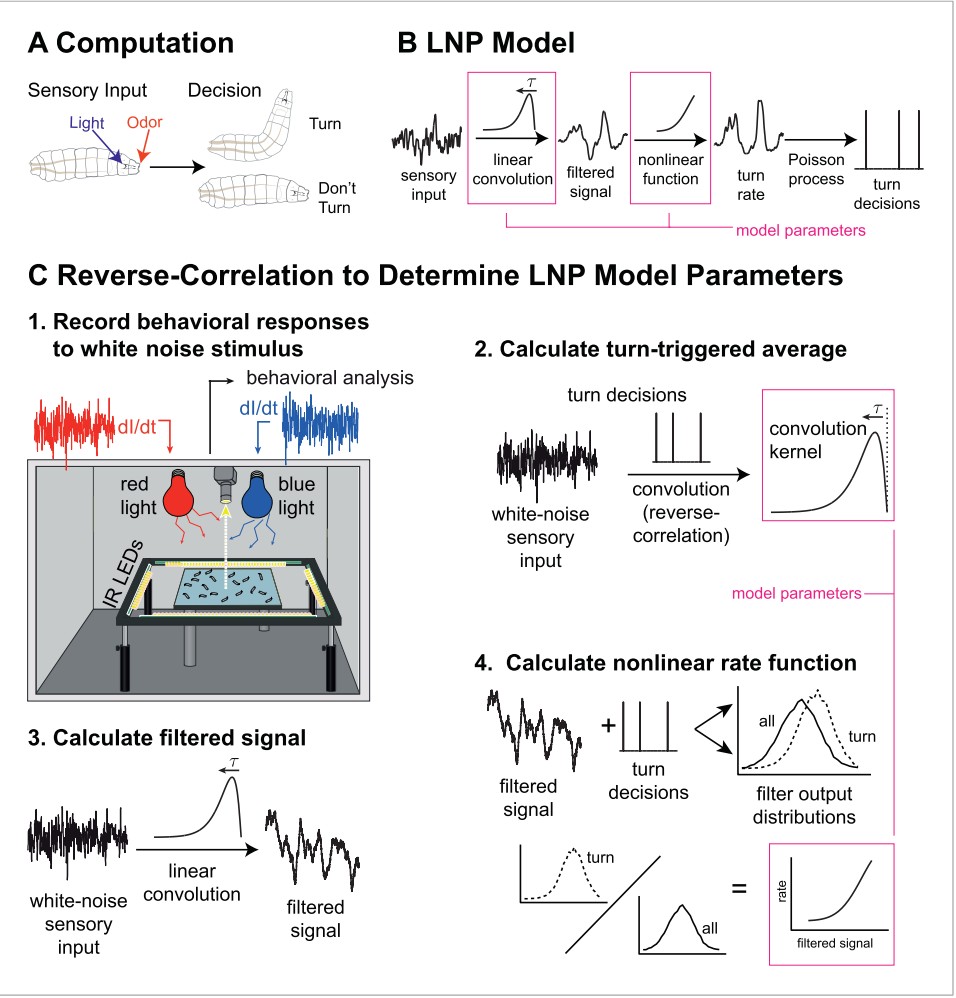

**Figure 1**. Identifying computations underlying the decision to initiate a turn. (**A**) Computation: on the basis of sensory input (light or odor in this work) the larva decides whether or not to end a run and begin a turn. (**B**) LNP model of the computation: Sensory input is processed by a linear filter to produce an intermediate signal. The rate at which the larva initiates turns is a nonlinear function of this signal. Turns are initiated stochastically according to a Poisson process with this underlying turn rate. (**C**) Reverse-correlation to determine LNP parameters: (**1**) We presented groups of larvae with either blue or red light with randomly varying intensity derivatives. Blue light provided a visual stimulus, while red light activated CsChrimson expressed in sensory neurons. In multi-sensory experiments, uncorrelated red light and blue light signals were presented simultaneously. Larvae were observed under infrared illumination, and their behaviors were analyzed with machine vision software. (**2**) We calculated the 'turn-triggered average' (TTA) or the reverse-correlation between turn initiation and stimulus by averaging the stimulus that preceded each moment any larva initiated a turn. The TTA approximates the convolution kernel for the linear response of the LNP model. (**3**) Using the TTA as a convolution kernel and the known input signal, we computed the intermediate filtered signal. (**4**) Using the inferred filtered signal and the observed times at which turns were initiated, we found the nonlinear rate function by dividing the distribution of filtered signal values at the time of turn initiation ('turn-triggered-ensemble') by the distribution of all filtered signal values. Illustrations adapted from (**Kane et al., 2013**).

where $S(t)$ is the input signal, $A$ is a linear filter, and $f$ is a static nonlinear function (**Figure 1B**). Thus for uni-modal inputs, we aimed to develop models of the form:

$$r_O(t) = f(x_O(t)) \; (\text{odor}),$$

$$r_L(t) = g(x_L(t)) \; (\text{light}),$$

where $x_O(t) = (A_O * S_O)(t)$ and $x_L(t) = (A_L * S_L)(t)$ are the outputs of the linear filters for odor and for light, respectively. For multi-modal input, we sought either a model in which turns are initiated independently by separate circuits

$$r_{O-L}(t) = f(x_O(t)) + g(x_L(t)) \text{ (independent pathways)},$$

or one in which odor and light information are combined more generally

$$r_{O-L}(t) = h(x_O(t), x_L(t)) \text{ (nonlinear integration)}.$$

## Results

### Reverse-correlation with visual and fictive olfactory stimuli

We began by exploring larvae's responses to uni-modal stimuli. *Drosophila* larvae avoid light and carbon dioxide and are attracted to Ethyl Acetate (EtAc). The sensory input pathways are well-characterized for each of these stimuli. The larva's navigational response to light is mediated primarily by four photoreceptor neurons in each of two primitive eye-spots (*Hassan et al., 2005*; *Sprecher and Desplan, 2008*; *Keene et al., 2011*; *Keene and Sprecher, 2012*; *Kane et al., 2013*). A single pair of Gr21a expressing receptor neurons mediates the larva's $CO_2$ response (*Python and Stocker, 2002*; *Faucher, 2006*; *Jones et al., 2007*; *Kwon et al., 2007*). Or42a and Or42b are the primary EtAc olfactory receptors (*Kreher et al., 2005*, *2008*; *Asahina et al., 2009*), and larvae are capable of decoding odor gradients on the basis of only Or42a or Or42b receptor neurons (*Louis et al., 2008*; *Asahina et al., 2009*).

Larvae initiate turns in response to increasing light intensities (*Sawin et al., 1994*; *Hassan et al., 2005*; *Scantlebury et al., 2007*; *Kane et al., 2013*) and carbon dioxide concentrations (*Gershow et al., 2012*) and decreasing EtAc concentrations (*Gomez-Marin et al., 2011*; *Gershow et al., 2012*). To investigate the larva's decision to turn in response to visual cues, we presented 448 nm blue light to wild-type larvae. To probe olfactory computations, we expressed *UAS-CsChrimson* (*Klapoetke et al., 2014*), a red light activable cation channel, in Gr21a, Or42a, and Or42b receptor neuron pairs. We used 655 nm red light (outside the sensitive range of the larva's visual pigments [*Salcedo et al., 1999*]) to activate these neurons while presenting a constant dim blue light background (*Klapoetke et al., 2014*) to mask any visual response to the red light.

We presented groups of larvae with fluctuating levels of red or blue light and analyzed their resulting behaviors using machine vision software (*Gershow et al., 2012*) to identify each navigational decision (*Figure 1C-1*). We determined the parameters of the LNP model by measuring the reverse-correlation (*Chichilnisky, 2001*; *Dayan, 2001*; *Westwick et al., 2003*; *Ringach and Shapley, 2004*; *Bialek and van Steveninck, 2005*; *Schwartz et al., 2006*; *Kim et al., 2011*; *Klein et al., 2014*, *Theobald et al., 2010*) between the stimulus and evoked behaviors (*Figure 1C, Video 1*).

For light, odor, and carbon dioxide, the derivative of stimulus intensity is more salient to larvae than the stimulus value itself (*Gomez-Marin et al., 2011*; *Gershow et al., 2012*;

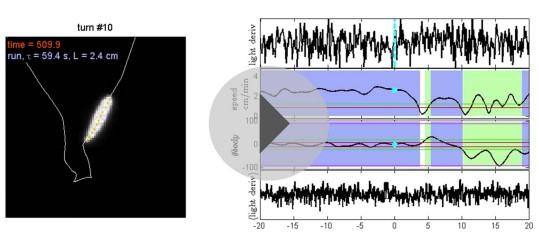

**Video 1.** Calculating the turn-triggered average. **Left panel:** annotated video image of individual larva. Thin white line: larva's path (past and future). Gold dots: markers along midline of animal, used to determine posture. Upper left corner: time (since experiment start) and behavioral state. **Right panels:** top: light derivative (AU) vs. time; current time is indicated by dashed cyan line. Middle panels: speed and body bend angle, metrics used to determine behavioral state, vs. time. Shading indicates behavioral state (blue = run, white = turn; within turns, red = rejected head sweep, green = accepted head sweep). Current time is indicated by cyan dot. Bottom panel: turn-triggered average light (TTA) intensity derivative (AU), calculated based on turns preceding the one shown. **Animation:** The time preceding and following individual turns is featured. At the moment a larva initiates a turn, we 'grab' the sequence of light intensity derivatives and add it to a running average (shown at the bottom). As we include more turns in the average (number of included turns is indicated by 'turn #' above the left panel—note logarithmic spacing of turn #s), we build up a 'TTA' that approximates the linear filter in the LNP model.

*Kane et al., 2013*; *Gomez-Marin and Louis, 2014*). Typically in reverse-correlation experiments, rapidly changing uncorrelated stimulus intensities are provided (e.g. by choosing random binary values, random normally distributed values, or values from an M-sequence). Because of regression to the mean, these sequences have derivatives that are anticorrelated on time scales longer than the update period. Thus, if we provided uncorrelated stimulus intensities and larvae turned in response to an increase in light, we would expect an average decrease in light after turns. This would complicate analyses of decisions, like whether to accept or reject a head sweep, the larva makes following a turn. We therefore chose a stimulus optimized for analysis of the larva's response to derivatives, a Brownian random walk, whose derivatives (on all time scales) are independent identically distributed Gaussian variables.

## Computations underlying the decision to turn in response to visual or olfactory stimuli

First, we computed the turn-triggered average (TTA) of the light intensity derivatives (*Figure 2A*). We found: wild-type larvae turn in response to an increase in blue light but were unresponsive to red light masked by dim blue light (even when fed all-trans-retinal); larvae expressing CsChrimson in EtAc sensing neurons (*Or42a>CsChrimson*, *Or42b>CsChrimson*) turned in response to a *decrease* in red light; and larvae expressing CsChrimson in the $CO_2$ receptor (*Gr21a>CsChrimson*) turned in response to an *increase* in red light. These results match the larva's strategies for navigating static gradients of natural stimuli; turning in response to increases in light, decreases in EtAc, and increases in $CO_2$. In all cases (excepting the wild-type red-light control), larvae considered changes over the previous ~2.5 s when deciding to turn, and derivatives were maximally salient about 0.6–0.8 s before a turn was actually initiated.

For each set of reverse-correlation experiments, we used the TTA as the convolution kernel to calculate the linear filter outputs and computed the turn rate as a nonlinear function of the filter output (*Figure 2B*). To test the predictive power of the LNP model, we presented larvae with light intensity square waves. We compared the resulting turn rates to those predicted by our LNP model fits (*Figure 2C*). The cyan lines in *Figure 2C* show the exact predictions of the model, with no free parameters. We found that in all cases, except for *Gr21a>CsChrimson*, the step response has a longer duration than predicted by the LNP model, and there is an unpredicted sustained decrease in turn rate following a favorable change (at t = 0), but given the inherent variability of behavior and the simplicity of the LNP model, the reverse-correlation experiments predicted the temporal course of the larva's response to step changes surprisingly well.

We were concerned that over the course of the 20 min behavioral experiments, continuous exposure to red light might exhaust the ability of CsChrimson to excite neural activity or that larvae might adapt to the stimulus presentation and cease responding. To test whether this was the case, we separately analyzed the first and second 10 min of the experiments (*Figure 2—figure supplement 1*). In all cases, we recovered the same TTA (*Figure 2—figure supplement 1A*) using data from only the first 10 min, only the second 10 min, or the entire data set. We found that for visual and attractive odor inputs, the slope of the nonlinear function was steeper (*Figure 2—figure supplement 1B*) in the first 10 min than in the second 10, indicating larvae were slightly less responsive to stimulus changes in the latter half of the experiments. For fictive $CO_2$, on the other hand, we found that larvae were actually more responsive in the second 10 min, representing sensitization rather than adaptation. In all cases, the results for the two halves of the experiments were similar enough to justify using all 20 min of data in our analyses. To test the self-consistency of our model, we used the parameters extracted from the first 10 min of experiments to predict the larva's turning rate in the second 10 min (*Figure 2—figure supplement 1C*). We found excellent agreement between predictions and measurements at lower turn rates. At higher turn rates, we found that in the second 10 min, larvae turned less than predicted for light and attractive odor cues and more than predicted for $CO_2$ cues, again reflecting modest adaptation and sensitization.

## Computations determining turn size and turn direction

Next, we explored how larvae bias the size and directions of turns. Larvae make larger turns when subjected to unfavorably changing conditions (*Luo et al., 2010*; *Gershow et al., 2012*; *Kane et al., 2013*), so we calculated the TTA for large and small turns separately (*Figure 2D*). For approximately

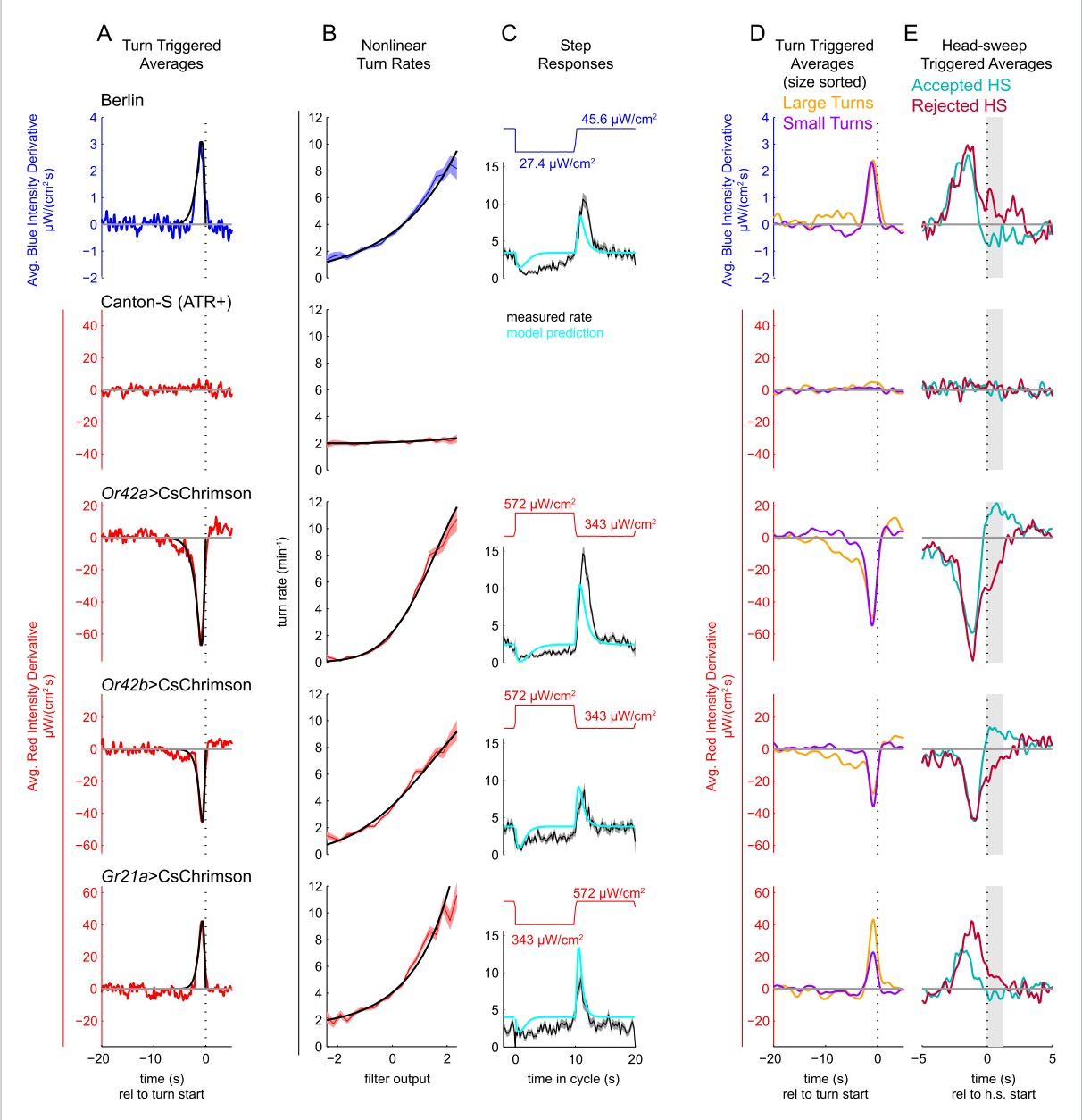

**Figure 2.** Unimodal reverse-correlation experiments. Top row, Berlin wild-type larvae were stimulated with blue ($\lambda_{peak}$ = 448 nm; max intensity = 74 μW/cm²) light. All other rows, larvae of indicated genotype were stimulated with red light ($\lambda_{peak}$ = 655 nm; max intensity = 911 μW/cm²) while constant dim blue light (intensity = 3.7 μW/cm²) served as a visual mask. Column **A**: Turn triggered average. Average stimulus preceding (and following) each turn initiation. Turns are initiated at time 0 (indicated with dashed line). The black line is the smoothed TTA used as the linear filter. Column **B**: Measured turn rates as a function of calculated filter output. Line and shaded region represent mean turn rate and standard error due to counting statistics. Black line is the nonlinear turn rate modeled as a ratio-of-Gaussians (**_Pillow and Simoncelli, 2006_**). Column **C**: Step responses predicted by LNP model. Square waves of light with period 20 s and duty cycle 50% were presented to larvae. The LNP model was used to predict the resulting turn rates. Top graphs: light level vs time in cycle. A favorable change happens at t = 0 and an unfavorable change at t = 10 s. Bottom graphs: measured and predicted turn rates vs time in cycle. Black line and shaded region represent mean turn rate and standard error due to counting statistics. The cyan line is the exact prediction of the model using the parameters found from the corresponding reverse-correlation experiments. (**A**, **B**) The stimulus and analysis were cyclic, so the time range from −2 to 0 s is identical to that from 18–20 s. Column **D**: Size-sorted turn-triggered average. As in **A**, but turns were sorted into large (heading change during turn > rms heading change) and small turns. Displayed averages are lowpassed with a Gaussian filter ($\sigma$ = 0.5 s) to clarify the long time-scale features. Column **E**: Head-sweep triggered

*Figure 2. continued on next page*

*Figure 2. Continued*

average (for first head-sweep of turn). Average stimulus surrounding accepted (teal) and rejected (red) head-sweeps. Head sweeps were initiated at time t = 0 and concluded at a variable time in the future. The mean head-sweep duration (1.25 s) is indicated by the shaded region. See *Table 1* for number of experiments, animals, and so on.

The following figure supplement is available for figure 2:

**Figure supplement 1**. LNP model parameters are stable for duration of 20 min experiments.

10–15 s prior to the start of large turns, there was an average gradual increase in blue light levels for visual experiments, and an average gradual decrease in red light levels for *Or42a>CsChrimson* and *Or42b>CsChrimson* experiments. Over the same time period, for small turns, there was a slight average decrease in blue light and increase in red light. Interestingly, immediately prior to the initiation of a turn, the average change in stimulus was the same for large and small turns. Thus, larvae consider the change in light intensity and attractive odor concentration over the previous 10–15 s when deciding the size of their turns, but the size of the stimulus change that leads the larva to actually initiate a turn appears not to influence turn size. In contrast, for *Gr21a>CsChrimson* larvae, the size of a turn was determined by the magnitude of the increase immediately preceding a turn.

After initiating turns, larvae use head-sweeps as probes to find a favorable direction for the next run (*Luo et al., 2010*; *Gomez-Marin et al., 2011*; *Gershow et al., 2012*; *Kane et al., 2013*; *Gomez-Marin and Louis, 2014*; *Klein et al., 2014*). Head-sweeps in favorable directions are more likely to be accepted, beginning a new run in that direction. Head-sweeps in unfavorable directions are more likely to be rejected, resulting in one or more additional head-sweeps. To characterize the larva's decision to accept or reject a head-sweep, we measured the head-sweep triggered average, aligned to the start of a head-sweep, separately for rejected and accepted head-sweeps (*Figure 2E*). All head-sweep triggered averages show a large change immediately before the start of the head-sweep; this is the stimulus change that triggered the larva's decision to turn. We found that during accepted head-sweeps, average blue light levels and Gr21a activity decreased and Or42a and Or42b activity increased, all favorable changes. During rejected head-sweeps, the reverse was true: average blue light and Gr21a activity levels increased and Or42a and Or42b activity decreased. Surprisingly, for *Gr21a>CsChrimson* larvae, larger increases in activity prior to the head-sweep start led to an increased probability of rejecting the head-sweep.

Our uni-modal experiments showed that CsChrimson induced activity in olfactory neurons evokes navigational behaviors and that reverse correlation can be used to identify both visual and olfactory computations. We also found that activity in $CO_2$ receptor neurons is interpreted according to different rules than for light or attractive odors.

## Odor-light integration—computations underlying the decision to turn

We next asked how the larva integrates visual and olfactory information when making navigational decisions. We carried out reverse-correlation experiments with simultaneous uncorrelated light and attractive odor stimuli, using dim blue light to activate the visual system and intense red light to activate CsChrimson expressed in *Or42a* receptor neurons. We found the TTA for both signals (*Figure 3A*), and we applied the resulting filters to our input signals to find $x_O(t)$ and $x_L(t)$, the outputs of the linear odor and light filters, at each point in time. We scaled the filters so that $x_O(t)$ and $x_L(t)$ had unit variance in the stimulus ensemble (*Pillow and Simoncelli, 2006*). To determine whether the larva's turning decisions result from independent olfactory and visual pathways, we examined the statistics of the *turn-triggered ensemble* (*Figure 3B*). In our white noise experiments, $x_O(t)$ and $x_L(t)$ are independent Gaussian variables with mean 0, so it can be shown (*Bialek and van Steveninck, 2005*)

$$E[x_O x_L | turn] \propto E\left[ \partial^2 r_{O-L}(t) / \partial x_O \partial x_L \right].$$

In general, this value will be nonzero, but in the independent pathways model (*Figure 3C*, *Figure 3—figure supplement 1*), the mixed partial derivative is identically 0, so

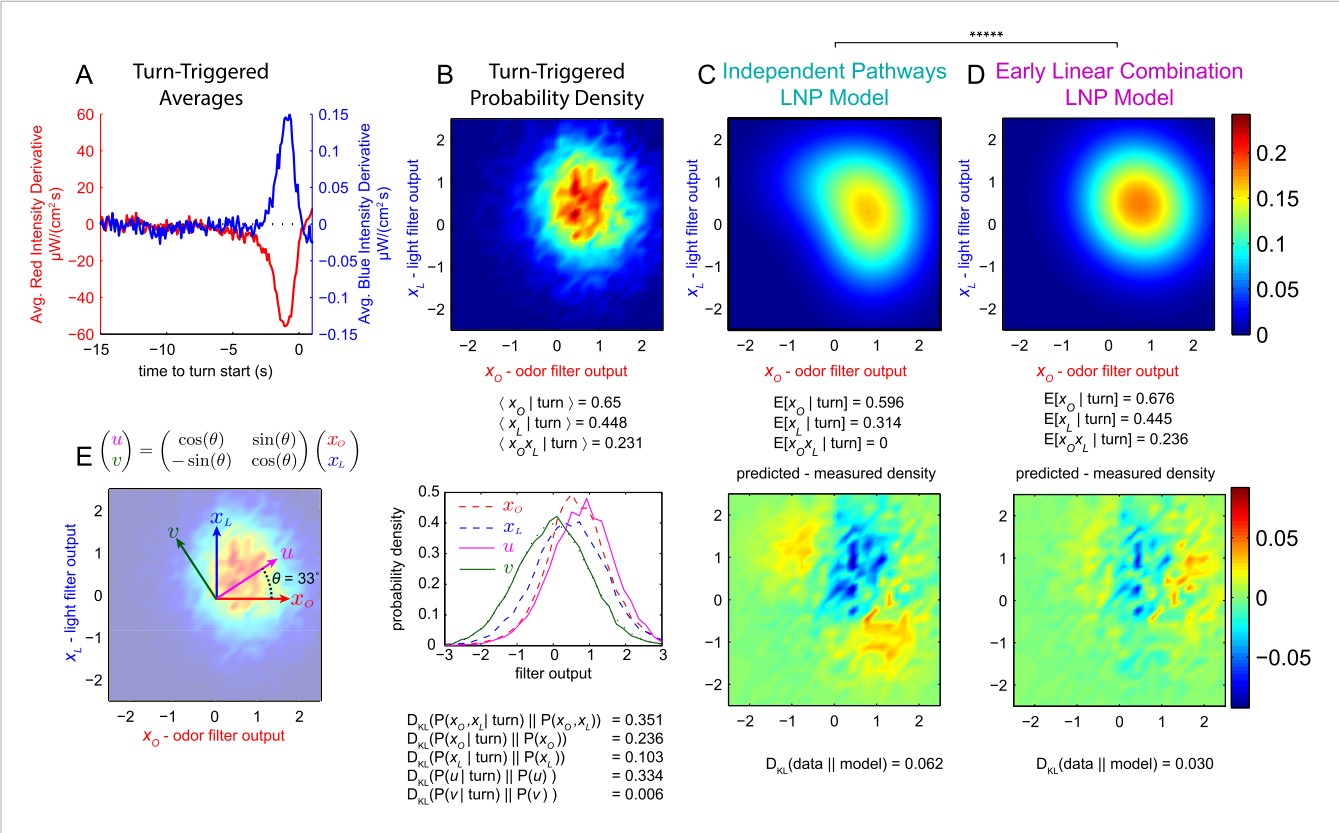

**Figure 3**. Multi-modal reverse-correlation experiments suggest attractive odor and light signals are combined linearly and early. *Or42a>CsChrimson* larvae were presented with independently varying Brownian light intensities. Reverse-correlation analysis was carried out as in *Figure 1*. (**A**) TTA. Average change in red (fictive odor) and blue light intensities preceding turns. (**B**) Turn triggered ensemble. Top: 2D density histogram of calculated odor and light filter outputs at initiation of each turn. Bottom: 1D density histograms of filter outputs ($x_O$, $x_L$) and their linear combinations ($u$, $v$). $D_{KL}(P(x|turn)||P(x))$ is the Kullback-Leibler divergence from the turn-triggered distribution to the distribution of $x$ at all times. Larger values indicate that $x$ carries more information about the decision to turn. (**C**, **D**) Predicted turn triggered ensemble according to, (**C**) independent pathways model and (**D**) early linear combination model. Top panel: predicted density. Bottom panel: difference between predicted density and measured density. $D_{KL}(data||model)$ is the Kullback-Leibler divergence of the model from the data; smaller values indicate a better match. ***** = P (Independent pathways model)/P (Early linear combination model) < 0.00001; Aikake Information Criterion Test. (**E**) Coordinate rotation described in text and used in bottom panel of **B**. Orthogonal coordinates ($u$, $v$) are rotated 33° relative to ($x_O$, $x_L$). Rotation is shown overlaid on turn-triggered probability density (**B**). See *Table 1* for number of experiments, animals, and so on.

The following figure supplements are available for figure 3:

**Figure supplement 1**. Graphical explanation of the independent pathways model.

**Figure supplement 2**. Graphical explanation of the early linear combination model.

**Figure supplement 3**. Visual and fictive olfactory stimuli do not cross-talk.

$$E[x_O x_L | turn] = 0 \text{ (independent pathways).}$$

In fact, we found $<x_O x_L | turn> = 0.23$, disfavoring the independent pathways hypothesis.

If larvae integrate odor and light information when making turning decisions, what form does this integration take? A potentially favorable (*Ma et al., 2006*; *Angelaki et al., 2009*) approach would be for the larva to use a simple linear combination of uni-modal filter outputs as the basis for downstream multi-modal processing (*Figure 3—figure supplement 2*). In this case, the turn rate would be given by

$$r_{O-L} = h(\cos(\theta)x_O(t) + \sin(\theta)x_L(t)) \text{ (early linear combination)}.$$

Here, $\theta$ is a constant reflecting the relative importance of each filter output to the computation ($\theta = 0$ would mean larvae respond only to odor and $\theta = 90°$ would mean larvae respond only to light).

We used both the independent pathways and early linear combination models to fit the turn-triggered probability density of ($x_O$, $x_L$) and found better agreement between the data and the early linear combination prediction (**Figure 3D**) than the independent pathways prediction (**Figure 3C**).

In the early linear combination model, the turn rate is a function of a 1-dimensional stimulus vector—$\cos(\theta)x_O(t) + \sin(\theta)x_L(t)$—so a rotation of odor-light convolution space (**Figure 3E**)

$$\begin{pmatrix} u(t) \\ v(t) \end{pmatrix} = \begin{pmatrix} \cos(\theta) & \sin(\theta) \\ -\sin(\theta) & \cos(\theta) \end{pmatrix} \begin{pmatrix} x_O(t) \\ x_L(t) \end{pmatrix},$$

should produce a single parameter, $u$, that carries as much information about turn decisions as the pair ($x_O$, $x_L$) together, and an orthogonal parameter, $v$, that carries no information at all. The Kullback-Leibler divergence between the stimulus and turn-triggered distributions describes the amount of information stimulus parameters carry about the decision to initiate a turn (**Pillow and Simoncelli, 2006**); we calculated this divergence for the pair ($x_O$, $x_L$) and for $x_O$, $x_L$, $u$, and $v$ individually (**Figure 3B**) and found that $u$ alone is nearly as informative as both $x_O$ and $x_L$ and that $v$ contributes very little to the decision to turn, further supporting the early linear combination model.

Can the observed summation of visual and odor inputs be explained by blue light activation of the CsChrimson channel? To probe for cross-talk between visual and olfactory channels, we repeated the multi-sensory experiments (again presenting both red and blue stimuli simultaneously) using genetically blinded larvae expressing CsChrimson in Or42a neurons and using wild-type larvae not expressing CsChrimson (**Figure 3—figure supplement 3**). Blind larvae responded only to red light (**Figure 3—figure supplement 3B**) while larvae not expressing CsChrimson responded only to blue light (**Figure 3—figure supplement 3C**).

To further compare the independent pathways and early linear combination models, we measured larvae's responses to simultaneous step changes in *Or42a* receptor neuron activity and blue light (**Figure 4**). We presented larvae with all possible combinations of favorable, neutral, and unfavorable steps of red light (fictive odor) and blue light (visual cue). We used the kernels calculated from the reverse-correlation experiments and fit the nonlinear functions parameterizing the early linear combination and independent pathways model to the observed turn rates. We found that despite having fewer free fit parameters, the early linear combination model (magenta line, **Figure 4**) better predicted the response to conflicting and aligned multi-sensory input than the independent pathways model (cyan line, **Figure 4**).

For changes in only light or only odor, larvae increase their turn rates significantly more for unfavorable changes (panels ii and iii) than they decrease their turning in response to favorable changes (i and vi). Thus the independent pathways model, which sums separate rates for light and odor changes, predicts that in response to conflicting favorable and unfavorable changes (iv and viii), the larvae will still significantly increase their turning. Instead, conflicting changes in odor and light cancel each other out, as predicted by the early linear combination model. That larvae turn in response to increases in blue light (iii, v) but decreases in red light (ii, v) further confirms that the two light sources are activating different sensory pathways. For instance, if blue light were primarily activating CsChrimson, we would expect that decreasing blue and red light together (viii) would provoke more turning than decreasing red light alone (ii), but in fact *increasing* blue light while *decreasing* red light (v) provokes the largest increase in turning.

## Odor-light integration—computations underlying turn size and turn direction

We wondered whether the larva might use the same linear combination of light and odor signals to make other navigational decisions, like whether to accept or reject head-sweeps. Although we previously determined a maximally informative combination of *filtered* odor and light inputs, the convolution kernels have similar shapes, so we might reasonably use the same rules to combine the raw input stimuli.

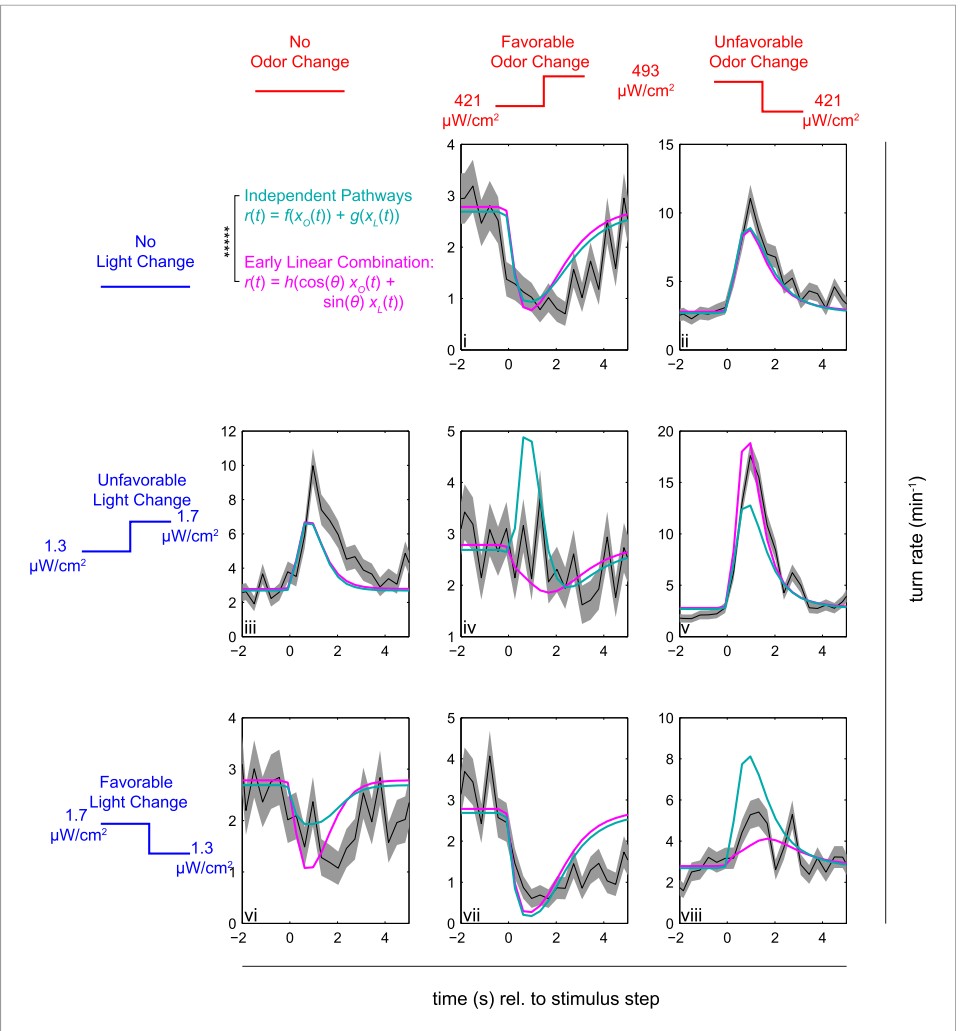

**Figure 4**. Multi-modal step responses support early linear combination of odor and light signals. Turn rates vs time for *Or42a>CsChrimson* larvae responding to coordinated increases and decreases of red and blue light. All steps occur at t = 0. Left column: no change in fictive odor, center column: red light increases at t = 0 right column: red light decreases at t = 0. Top row: no change in visual stimulus, center row: blue light increases at t = 0, bottom row: blue light decreases at t = 0. For instance, in panel iii blue light increased at time 0, while red light remained constant; in iv, both red and blue light increased at time 0; and in v, blue light increased and red light decreased at time 0. Black line and shaded region represent mean turn rate and standard error due to counting statistics. Cyan line is the best-fit (maximum likelihood estimate, 6 parameter fit) prediction of the independent pathways model. Magenta line is the best-fit (maximum likelihood estimate, 4 parameter fit) prediction of the early linear combination model. Note that the time axis is the same for each subplot, but the turn rate axis varies. ***** = P (Independent pathways model)/P (Early linear combination model) < 0.00001; Aikake Information Criterion Test, measured for the entire data set. See *Table 1* for number of experiments, animals, and so on.

$$\begin{pmatrix} \mu(t) \\ \nu(t) \end{pmatrix} = \begin{pmatrix} \cos(\theta) & \sin(\theta) \\ -\sin(\theta) & \cos(\theta) \end{pmatrix} \begin{pmatrix} O(t) \\ L(t) \end{pmatrix}; O(t) = -\frac{dI_{red}}{dt} \Big/ I_r^0; L(t) = \frac{dI_{blue}}{dt} \Big/ I_b^0 \, .$$

Here, $I_r^0$ and $I_b^0$ are the normalization factors required to make the convolution kernels have unit variance, and $\theta$ was determined as a parameter of the early linear combination model (*Figure 3D*). We carried out reverse-correlation analysis on these rotated coordinates. We found: the turn triggered average of $\nu(t)$ was nearly 0 (*Figure 5A*); the size-sorted TTA of $\nu(t)$ was nearly 0 and the same for both small and large turns (*Figure 5B*); and the head-sweep triggered average of $\nu(t)$ was nearly 0 and the

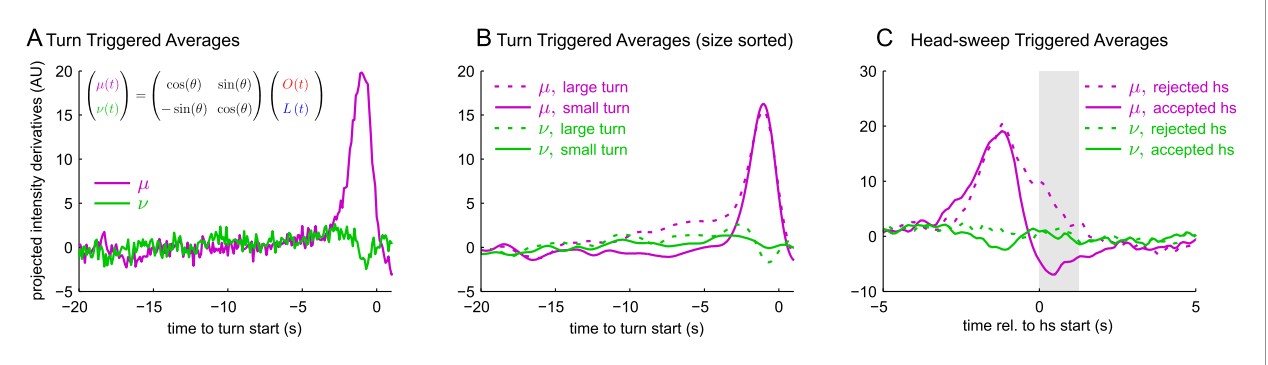

**Figure 5**. All navigational decisions appear to be based on a single linear combination of odor and light inputs. (**A–C**) Reverse correlation in rotated coordinate system. $\mu, \nu$ are linear combinations of the raw input stimuli according to the same scaling as used to combine filtered odor and light signals in *Figure 3*. (**A**) Turn-triggered averages. Average change in $\mu, \nu$ prior to start of a turn. (**B**) Size-sorted turn-triggered averages. Displayed averages are lowpassed with a Gaussian filter ($\sigma = 0.5$ s) to clarify the long time-scale features. (**C**) Head-sweep-triggered average (for first head-sweep of turn). Shaded region indicates mean head-sweep duration (1.25 s).

same for accepted and rejected head-sweeps (*Figure 5C*). Thus, larvae used a single linear combination of odor and light—$\mu(t)$—to determine whether to turn, how large of a turn to make, and whether to reject or accept head-sweeps, strongly suggesting odor and light inputs are combined at early stages of the navigational circuitry.

## Odor-light integration—probing for shifts in attention

Finally, we asked whether the larva might shift its attention between stimulus inputs. For instance, if a larva initiates a turn due to an increase in light, might it prioritize light changes over odor changes in deciding whether or not to accept the turn's first head-sweep? To test this, we re-examined the head-sweep acceptances and rejections from the multi-stimulus white noise experiments, sorting them based on the favorability of light and odor changes preceding the turn start (*Figure 6A*).

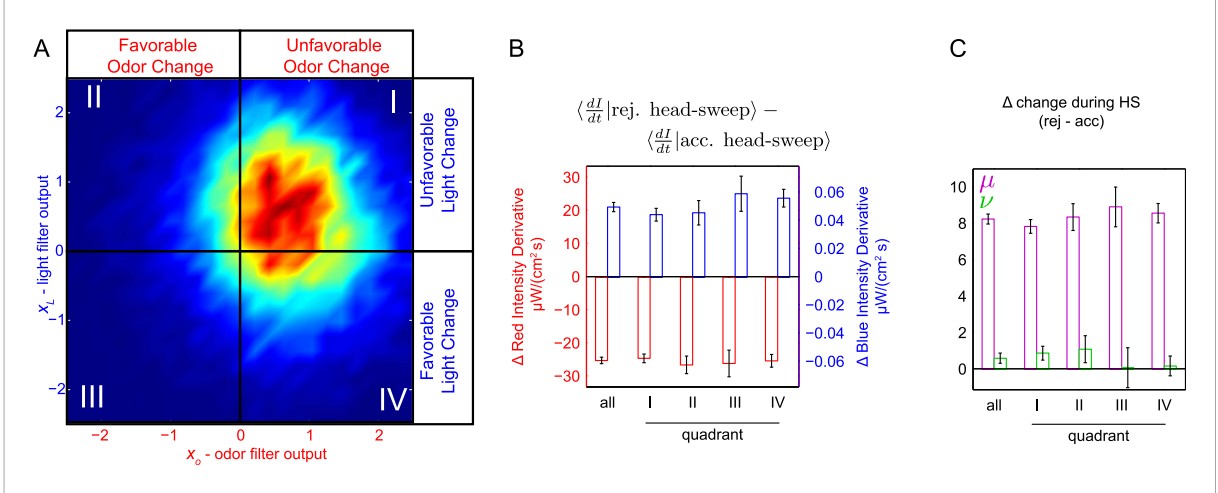

**Figure 6**. Probing for attentional shifts during multi-modal noise experiments. (**A**) Turn-triggered ensemble (duplicated from *Figure 3B*), with quadrants highlighted. Color scale the same as in 3B. Quadrants I–IV indicate which stimulus or stimuli likely provoked the larva to turn (I—both odor and light stimulated turning; II—light, but not odor, stimulated turning; III—neither odor nor light were changing unfavorably; IV—odor, but not light, stimulated turning). Each turn was assigned to one of these quadrants based on the filtered signal values ($x_O, x_L$) at the time the turn started. (**B**, **C**) Difference in intensity changes during accepted and rejected head sweeps (first head-sweep of turn only). Difference in mean rate of change in intensity over mean head sweep duration (1.25 s, shaded region in 5C) between rejected and accepted head-sweeps. Error bars are ±1 s.e.m. (**B**) Changes in odor and light intensities. (**C**) Changes in rotated coordinate system. See *Table 1* for number of experiments, animals, and so on.

For instance, in quadrant II, larvae turned following a recent favorable odor change and unfavorable light change. If an unfavorable light change caused otherwise neutral larvae to attend more strongly to light, we would expect that following a quadrant II turn, larvae would be more attentive to light changes than average. Alternately, if some larvae in the population were already attending to light more strongly than odor, we would expect them to be over-represented in the group of larvae turning in response to unfavorable changes in light coupled with favorable changes in odor. Thus, we would again expect that following a quadrant II turn, larvae would be attending more strongly to light than in the average population. Similarly, following quadrant IV turns, we would expect larvae to be attending more strongly to odor.

To measure the importance of light and odor changes during head sweeps, we subtracted the average derivative of light level during accepted head sweeps from the average derivative of light level during rejected head sweeps (*Figure 6B*). Regardless of which stimulus likely triggered the decision to turn, for both light and odor, we found the same differences between accepted and rejected head-sweeps. Thus, we found no evidence that the larva modulated the relative importance of light or odor changes when deciding whether to accept or reject a head-sweep. Indeed, for all combinations of light and odor changes preceding a turn, the same linear combination of odor and light inputs appeared to be equally salient in deciding whether to accept or reject a head-sweep (*Figure 6C*).

## Discussion

A key step in 'cracking' neural circuits is defining the computations carried out by those circuits (*Olsen and Wilson, 2008*; *Clark et al., 2013*). Recent work has refined the measurements of circuits' behavioral outputs, for example, from simply counting animals accumulating near an odor source (*Monte et al., 1989*) to specifying the sequence of motor outputs that allow odor gradient ascent (*Gomez-Marin et al., 2011*; *Gershow et al., 2012*; *Gomez-Marin and Louis, 2014*). Here, we carried on this refinement, quantifying the transformation from sensory activity to motor decision with sub-second temporal resolution. Our reverse-correlation analysis captured the essential features of the larva's navigational decision making, including the time scales and stimulus features associated with various decisions (*Figure 2*).

Our results are consistent with the understanding of how larvae navigate natural environments previously developed by observing behavior in structured environments of light or gaseous odors. For instance, when placed in environments with spatially varying Ethyl Butyrate (*Gomez-Marin et al., 2011*; *Gomez-Marin and Louis, 2014*) or Ethyl Acetate (*Gershow et al., 2012*) concentrations, larvae initiate turns more frequently when headed in directions of decreasing concentrations of these attractive odors. In this work, we showed that larvae initiate turns in response to a decrease in activity in Or42a or Or42b receptor neurons, the primary receptors for Ethyl Acetate and Ethyl Butyrate. Additionally, we showed that larvae mainly use only the previous two seconds of receptor activity to decide whether to turn (*Figure 2A*). This detail cannot be resolved from experiments in natural odor gradients, nor can the fact that larvae integrate changes in odor receptor activity over a much longer time period to decide the size of their turns (*Figure 2D*).

When larvae move their heads through a spatially heterogeneous environment, they generate changes in sensory input that could be used to decode local spatial gradients. It has been directly shown that warming a cold larva during a head-sweep causes the larva to accept that head-sweep, beginning a new run (*Luo et al., 2010*). For light and odor, a strong circumstantial case has been made that larvae use information gathered during head-sweeps to bias turn direction: the first head-sweep of a turn is unbiased (*Gershow et al., 2012*; *Kane et al., 2013*) but larvae are more likely to begin a run following a sweep in a direction of higher concentration of attractive odor, lower concentration of carbon dioxide, or lower luminosity (*Gomez-Marin et al., 2011*; *Gershow et al., 2012*; *Kane et al., 2013*; *Gomez-Marin and Louis, 2014*) and larvae with only a single functional odor receptor can still bias turn direction via head-sweeping (*Gomez-Marin and Louis, 2014*). In this work, we directly show that larvae do in fact use changes in odor receptor activity and light level measured during head-sweeps to determine whether to begin a new run or initiate a second head-sweep (*Figure 2E*).

Our experiments with a single stimulus also found a previously unknown difference in how larvae use $CO_2$ receptor activity to modulate turn size and head-sweep acceptance compared to visual

stimuli and to attractive olfactory receptor activity (*Figure 2D,E*). This could be related to a difference in how larvae modulate their forward motion in response to changes in $CO_2$ concentration compared to changes in light intensity or EtAc concentration. We previously measured larvae's responses to linear temporal ramps of light intensity (*Kane et al., 2013*), and EtAc and $CO_2$ concentrations (*Gershow et al., 2012*). For light and EtAc, larvae changed their rate of turning and the size of their turns in response to changing environmental conditions, but when they were actually engaged in forward movement, their speed of progress was the same whether conditions were improving or declining. In contrast, larvae dramatically decreased their forward run speed in response to increases in the concentration of $CO_2$. Larvae with non-functional $CO_2$ receptors did not change their speed at all in response to $CO_2$, so this modulation was due to a sensory-motor transformation and not to metabolic effects.

Larvae move forward through a series of tail to head peristaltic waves of muscle contraction (*Lahiri et al., 2011*; *Heckscher et al., 2012*) and modulate their forward speed by changing the frequency with which they initiate these waves (*Heckscher et al., 2012*). The observed $CO_2$ dependent speed modulation might therefore reflect the presence of a pathway by which Gr21a receptor neuron activity can down-regulate the probability of initiating forward peristaltic waves. In order to accept a head-sweep, that is, transition from head-sweeping to forward movement, larvae must initiate a new peristaltic wave (*Lahiri et al., 2011*). If an increase in Gr21a activity decreases the probability of initiating such a wave, this would explain why an increase in Gr21a activity *prior* to head-sweep initiation results in an increased probability of head-sweep rejection (*Figure 2E*). Similarly, an increase in Gr21a activity might bias the larva towards larger reorientations (*Figure 2D*) by decreasing the probability of quick, small course corrections.

In addition to defining the computations by which larvae navigate environments of varying light or varying odor, we also developed a quantitative model of odor-light integration. Previously, it has proven difficult to establish even a qualitative understanding of odor and light integration using static combinations of the two cues. Consider a simple experiment where a petri dish is divided into light and dark halves and a droplet of attractive odor is placed on the lighted half. If a larva moves towards the odor at the expense of moving out of darkness, is this because the larva naturally places more importance on odor than light regardless of intensity, because the particular concentrations of odor and intensities of light in the experiment favor a move towards odor, or because behavior is variable and larvae often make idiosyncratic choices? In our reverse-correlation experiments, we presented hundreds of larvae with thousands of combinations of light and odor variation and were thus able to resolve these ambiguities. We determined not just how larvae balance an overall attraction to odor and aversion to light, but how they combine transient odor and light signals to make individual navigational decisions.

## Conclusion

Here, we demonstrated the power of reverse-correlation analysis of larvae's behavioral responses to white-noise visual and fictive olfactory stimuli to decode the computations underlying the *Drosophila* larva's navigation of natural environments. We showed that this analysis could be used to decode the rules by which the larva integrates signals from distinct sensory organs. Larvae appear to use a single linear combination of odor and light inputs to make all navigational decisions, suggesting these signals are combined at early stages of the navigational circuitry.

In this work, we used optogenetics to explore how *perturbations in the activities* of identified neurons are interpreted behaviorally. We expressed CsChrimson in specific neurons to relate patterns of activity in these neurons to decisions regulating the frequency, size, and direction of turns (*Figure 2*). Using model parameters extracted from reverse-correlation experiments, we were able to predict how larvae would respond to novel perturbations of these neurons' activities (*Figure 2C*). We explored how activity in one particular neuron type modulated the larva's responses to a natural light stimulus (*Figures 3, 5, 6*) and predicted how the larva's natural response to blue light steps (*Figure 4-iii*) would be altered by simultaneous perturbation of this neuron (*Figure 4-iv,v*). Here we addressed sensory neurons, but our approach can be used generally to identify computations carried out on activities of interneurons, to determine whether activity in a neuron is interpreted as attractive or aversive, to measure how that activity combines

with other sources of information to produce decisions, and to find neurons most responsible for making navigational decisions (*Koulakov et al., 2005*).

## Materials and methods

### Fly strains
The following strains were used: Canton-S and Berlin wild type (gift of Justin Blau), $w^{1118}$;; *20XUAS-CsChrimson-mVenus* (Bloomington Stock #55136, gift of Vivek Jayaraman and Julie Simpson, Janelia Research Campus), *w\*;;Gr21a-Gal4* (Bloomington stock #23890), *w\*;;Or42a-Gal4* (Bloomington stock #9969), *w\*;;Or42b-Gal4* (Bloomington stock #9972), *GMR-hid/CyO,P{sevRas1.V12}* (Bloomington stock #5771).

### Crosses
40 virgin female *UAS-CsChrimson* flies were crossed with 20 males of the selected Gal4 line. F1 progeny of both sexes were used for experiments.

### Larva collection
Flies were placed in 60 mm embryo-collection cages (59–100, Genesee Scientific, San Diego, CA) and allowed to lay eggs for 3 hr at 25°C on enriched food media ('Nutri-Fly German Food', Genesee Scientific). For all experiments except the Berlin response to blue light (*Figure 2*, top row and *Figure 2—figure supplement 1*, top row), the food was supplemented with 0.1 mM all-trans-retinal (ATR, R2500, Sigma Aldrich, St. Louis, MO), and cages were kept in the dark during egg laying. When eggs were not being collected for experiments, flies were kept either on plain food or agar (neither containing ATR).

Petri dishes containing eggs and larvae were kept at 25°C (ATR+ plates were wrapped in foil) for 48–60 hr. Second instar larvae were separated from the food using 30% sucrose solution and washed in deionized water. Larval stage was verified by size and spiracle morphology. Preparations for experiments were carried out in a dark room, under dim red (for photo-taxis experiments) or blue (for CsChrimson experiments) illumination. Prior to beginning experiments, larvae were dark adapted on a clean 2.5% agar surface for a minimum of 10 min.

### Behavioral experiments
Approximately 30–50 larvae were transferred with a wet paintbrush to a 23 cm square dish (Corning BioAssay Dish #431111, Fisher Scientific, Pittsburgh, PA), containing 2.5% (wt/vol) bacteriological grade agar (Apex, cat#20-274, Genesee Scientific) and 0.75% (wt/vol) activated charcoal (DARCO G-60, Fisher Scientific). The charcoal darkened the agar and improved contrast in our dark-field imaging setup. The plate was placed in a darkened enclosure and larvae were observed under strobed 850 nm infrared illumination (ODL-300-850, Smart Vision Lights, Muskegon, MI) using a 14 fps 5 MP rolling shutter CMOS camera (Basler acA2500-14gm, Graftek Imaging, Austin, TX) in global-reset-release mode and an 18 mm c-mount lens (54-857, Edmund Optics, Barrington, NJ) equipped with an IR-pass filter (Hoya R-72, Edmund Optics). The experiments of *Figure 3—figure supplement 3B* were recorded using a 4 MP global shutter CMOS camera (Basler acA2040-90umNIR, Graftek Imaging) operating at 20 fps and a 35 mm focal length lens (Fujinon CF35HA-1, B&H Photo, New York, NY). A microcontroller (Teensy++ 2.0, PJRC, Sherwood, OR) coordinated the infrared strobe and control of the stimulus light source, so stimulus presentation and images could be aligned to within the width of the strobe window (2–5 ms). Videos were recorded using custom software written in LABVIEW and analyzed using the MAGAT analyzer software package (*Gershow et al., 2012*). Further analysis was carried out using custom MATLAB scripts. *Table 1* gives the number of experiments, animals, turns, and head sweeps analyzed for each experimental condition. Software is available at https://github.com/GershowLab.

### Stimulus light source
We built a custom circuit board (Advanced Circuits, Colorado) containing 66 deep red high brightness LEDs (Philips Lumileds, LXM3-PD01, 655 nm central wavelength) and 12 royal blue high brightness LEDs (LXML-PR01-0500, 447.5 nm central wavelength) evenly distributed

**Table 1.** Numbers of experiments, animals, turns, and head sweeps for all figures

| Genotype | #expts | #animals | hours | #turns | rms turn size | #large turns | #small turns | #accepted head sweeps | #rejected head sweeps |
|---|---|---|---|---|---|---|---|---|---|
| Uni-modal reverse-correlation experiments (*Figure 2A,B,D,E*, *Figure 2—figure supplement 1*) | | | | | | | | | |
| Berlin | 6 | 150 | 52.6 | 6594 | 76.4 | 2462 | 4132 | 4139 | 2455 |
| Canton-S | 7 | 334 | 117.7 | 8824 | 66.0 | 3086 | 5738 | 5570 | 3254 |
| Or42a>CsChrimson | 5 | 180 | 61.1 | 6971 | 68.9 | 2531 | 4440 | 4122 | 2849 |
| Or42b>CsChrimson | 6 | 246 | 64.4 | 9565 | 73.3 | 3480 | 6085 | 6215 | 3350 |
| Gr21a>CsChrimson | 5 | 227 | 54.2 | 8760 | 75.1 | 3392 | 5368 | 5424 | 3336 |
| Uni-modal step experiments (*Figure 2C*) | | | | | | | | | |
| Berlin | 4 | 95 | 34.7 | 3674 | – | – | – | – | – |
| Or42a>CsChrimson | 2 | 107 | 36.6 | 3905 | – | – | – | – | – |
| Or42b>CsChrimson | 2 | 99 | 21.5 | 2599 | – | – | – | – | – |
| Gr21a>CsChrimson | 2 | 111 | 22.1 | 2384 | – | – | – | – | – |
| Multi-modal reverse-correlation experiments (*Figure 3, 5, 6*, *Figure 3—figure supplement 3*) | | | | | | | | | |
| Or42a>CsChrimson | 12 | 608 | 136 | 21,075 | 66.6 | 7225 | 13,850 | 12,795 | 8280 |
| quadrant I | – | – | – | 10,363 | – | – | – | 6216 | 4147 |
| quadrant II | – | – | – | 3301 | – | – | – | 2086 | 1215 |
| quadrant III | – | – | – | 1684 | – | – | – | 1088 | 596 |
| quadrant IV | – | – | – | 5727 | – | – | – | 3405 | 2322 |
| GMR-Hid, Or42a>CsChrimson | 3 | 121 | 28.9 | 4842 | – | – | – | – | – |
| Canton-S | 3 | 166 | 39.9 | 3412 | – | – | – | – | – |
| Multi-modal step experiments (*Figure 4*) | | | | | | | | | |
| Or42a>CsChrimson | 5 | 250 | 50 | 7859 | – | – | – | – | – |

#expts: Number of 20 min experiments. For reverse-correlation experiments, each experiment presented a different stimulus sequence with the same statistical properties; for step experiments, the same stimulus pattern was presented in each experiment.

#animals: Approximate number of animals, taken by finding the maximum number of animals tracked in a 30-s window during each experiment.

#hours: total observation time in units of larva-hours. Observing 3 larvae for 20 min each would equal 1 larva-hour.

#turns: total number of turns observed and used in analysis.

rms turn size: root mean square turn size in degrees (defined as angular difference in run heading immediately before and after a turn) for the set of experiments.

#large/small turns: number of turns with angular changes larger/smaller than the rms turn size.

#accepted head sweeps: number of times the first head sweep of a turn was accepted, ending in a new run.

#rejected head sweeps: number of times the first head sweep of a turn was rejected, leading to another head sweep.

over ~25 cm × 25 cm. The LEDs were driven at constant current by a switch-mode LED driver circuit (based on LT3518, linear technology) operating at a switching frequency of 2 MHz. The on-current was set by interchangeable feedback resistors and could be modulated separately for red and blue LEDs. The intensity of the red and blue LEDs was controlled separately by pulse-width-modulation. Illumination was provided from above the larvae; the LED circuit board was at the same height as the recording camera (~50 cm above the behavioral arena).

For multi-sensory experiments, the maximum red light intensity (911 $\mu$W/cm$^2$) was 300 times greater than the maximum blue light intensity (3 $\mu$W/cm$^2$). CsChrimson is slightly more sensitive to 655 nm than 448 nm light, so the blue light signal perturbed olfactory receptor neuron activity by less than 0.3% of the red-light signal's perturbation.

We calibrated the optical power of the LEDs using a photodiode power meter (S121C, Thorlabs, New Jersey) set to the central wavelength of the LEDs. We measured the uniformity of the stimulus light sources by imaging a Lambertian projector screen (Dalite 41466, Cousin's Video, Ohio) placed in the plane of the experimental arena under stimulus LED illumination.

**Table 2**. Kullback-Leibler divergences for *Figure 3*

| KL divergence | k-NN | model data as normally distributed | Szegö-PSD method |
|---|---|---|---|
| *Figure 3B*: $D_{KL}((P(x_o, x_l|turn)\|P(x_o, x_L))$ | 0.351 | 0.325 | – |
| *Figure 3B*: $D_{KL}((P(x_o|turn)\|P(x_o))$ | 0.236 | 0.223 | 0.235 |
| *Figure 3B*: $D_{KL}((P(x_L|turn)\|P(x_L))$ | 0.103 | 0.100 | 0.104 |
| *Figure 3B*: $D_{KL}((P(u|turn)\|P(u))$ | 0.334 | 0.324 | 0.333 |
| *Figure 3B*: $D_{KL}((P(v|turn)\|P(v))$ | 0.006 | 0.0002 | 0.003 |
| *Figure 3C*: $D_{KL}(data\|model)$ | 0.062 | 0.035 | – |
| *Figure 3D*: $D_{KL}(data\|model)$ | 0.030 | 0.007 | – |

KL divergence: the divergence to be calculated. k-NN: divergence calculated using the k-nearest neighbors algorithm. This value is displayed in *Figure 3*. model data as normal distributed: the distributions are modeled as Gaussians, whose divergence is calculated analytically. Szegö-PSD method: divergence between 1D distributions calculated by an alternate method.

## Stimulus sequences

Stimulus protocols were generated with MATLAB and stored on an SD card for use by the microcontroller. Light intensity was modulated using pulse-width-modulation with a frequency of ~112 Hz (constrained to update exactly 8 eight times per camera frame).

## Brownian light intensity

We chose a Brownian random walk, whose derivatives on all time scales are independent identically distributed Gaussian variables, to analyze the larva's response to derivatives of stimulus intensity. Light levels were specified by values between 0 (off) and 255 (maximum intensity). Sequences of light levels corresponding to a random walk with reflecting boundary conditions were generated according to these rules:

$$I_0 = 127,$$

$$I_j = -I_j \ if \ I_j < 0,$$

$$I_j = 510 - I_j \ if \ I_j > 255,$$

$$I_{j+1} = I_j + N(0, \sigma),$$

where $N(0,\sigma)$ was a Gaussian random variable with mean 0 and variance $\sigma^2$. For the experiments described in this work $\sigma = 3$. At an update rate of 112 Hz, this represents a diffusion constant of 504 (light levels)$^2$/s. After the sequence of light levels was generated, the levels were rounded to the nearest integer value before being transferred to the microcontroller for use in experiments. Sequences were not reused within an experimental group but might be reused between groups. For multi-modal experiments, independent sequences were used for each stimulus.

## Step responses

For step response experiments of *Figure 2C*, a square wave with a period of 20 s and duty cycle 50% (10 s high, 10 low) was presented. The low and high intensities were symmetrically distributed about the mean light intensity of the reverse-correlation experiments.

For the coordinated step response experiments of *Figure 4*, we presented steps of red and blue light intensity. Every 10 s, each signal either increased from low to high, decreased from high to low, or remained constant. The sequence of steps was chosen so that all combinations (except for both levels remaining constant) were presented. For each stimulus, the low and high intensities were symmetrically distributed about the mean light intensity of the reverse-correlation experiments.

## Data analysis

As described previously (**Gershow et al., 2012**; **Kane et al., 2013**), videos of behaving larvae were recorded using LabView software into a compressed image format (mmf) that discards the stationary background. These videos were processed using computer vision software (written in C++ using the openCV library) to find the position and posture (head, tail, midpoint, and midline) of each larva and to assemble these into tracks, each following the movement of a single larva through time. These tracks were analyzed by Matlab software to identify behaviors, especially runs, turns, and head sweeps.

The sequence of light intensities presented to the larvae was stored with the video recordings and used for reverse-correlation analysis.

## Reverse correlation analysis

### Turn-triggered averages

Turn-triggered averages (**Figures 2A, 3A, 5A**) with a bin size of 0.1 s were computed by averaging stimulus values at the corresponding times relative to the start of a turn; for example, the TTA at −1 s represents the average value of the derivative of the light intensity at all times that were between −0.95 and −1.05 s before the initiation of a turn. The TTA at +1 s represents the same average for times 0.95 to 1.05 s after the initiation of a turn. We computed the TTA at positive times as a control—we expected the average to be nearly 0 at positive times because behavior is causal (decisions contain no information about the future stimulus). Due to the reflecting boundary conditions, on long time scales, derivatives of the stimulus were anti-correlated; thus the TTA is not exactly 0 at positive times.

### Convolution kernels

We smoothed the TTA by fitting it to the impulse response of a third order linear system used to describe the calcium dynamics of *Caenorhabditis elegans* olfactory neurons (**Kato et al., 2014**). This fit was used only to smooth the TTA; we do not ascribe any biological significance to the fit parameters. The smoothed kernel is shown as a black line in **Figure 2A**.

### Filtered stimulus

We used the smoothed TTA as a convolution kernel to find the output of the linear filter stage of the LNP model. We scaled the kernel so that the variance of the filtered signal over the entire stimulus history was 1. For the Canton-S red light experiments (**Figure 2**, second row), a kernel could not be recovered from the TTA, so the kernel for Gr21a>CsChrimson was used to calculate the turn-rate.

### Nonlinear turn rates

(**Figure 2B**) The turn rate $r(x_f)$ and standard error $\sigma_r(x_f)$ as a function of filtered signal value ($x_f$) were computed by

$$r(x_f) = \frac{N_{turn}(x_f)}{N_{all}(x_f)} * \frac{1}{\Delta t}.$$

$$\sigma_r(x_f) = \frac{\sqrt{N_{turn}(x_f)}}{N_{all}(x_f)} * \frac{1}{\Delta t}.$$

$N_{turn}$ is the number of turns observed with the filtered signal within the histogram bin (size = 0.25) containing $x_f$ and $N_{all}$ is the total number of data points where the filtered signal was in the histogram bin and larvae were in runs and thus capable of initiating turns, and $\Delta t$ was the sampling period (1/14 s).

By construction, the stimulus ensemble is Gaussian distributed with mean 0 and variance 1. If the turn-triggered ensemble is also Gaussian distributed, then the turn rate is given by a ratio-of-Gaussians (**Pillow and Simoncelli, 2006**)

$$r_{ROG}(x) = \bar{r} \frac{e^{-\frac{(x-\mu)^2}{2\sigma^2}}}{\sigma e^{-\frac{x^2}{2}}}; \bar{r} = \frac{N_{turn}}{T}; \mu = E[x_f|turn]; \sigma^2 = E\left[(x_f - \mu)^2|turn\right].$$

T is the total time the larvae were in runs and therefore able to initiate turns. $r_{ROG}(x)$ is plotted in **Figure 2B** as a black line. This rate function was used for the predictions in **Figure 2C** (cyan lines), with $\bar{r}, \sigma, and \mu$ calculated directly from the turn-triggered ensemble.

## Size-sorted TTAs

Size-sorted TTAs (*Figures 2D, 5B*) were computed in the same manner, but the analysis was conducted separately for turns where the resulting heading change (difference in headings between end of the previous and beginning of the next runs) was larger or smaller than the rms heading change for the population over the course of the experiments. The turn-size is determined by the size of the initial head-sweep of the turn and by decisions made after turn-initiation (at positive times), so we did not expect the size-sorted TTA to be identically 0 at positive times.

## Head sweep triggered averages

(*Figures 2E, 5C*) As for the TTA, but the reference time (0) was chosen as the beginning of either rejected or accepted head sweeps. Because the decision to accept or reject a head sweep is made after the beginning of the head sweep, we expected that the average would be nonzero at positive times corresponding to the duration of a head sweep. To simplify interpretation of the resulting averages, we considered only the first head sweep of each turn.

## Maximum-likelihood estimation of turn-rate parameters

In *Figures 3, 4*, we fit the nonlinear turn rate to the observed data using the ratio-of-Gaussians function with $\bar{r}, \sigma, and\ \mu$ as fit parameters. The probability of observing at least one turn in an interval $\Delta t$ given an underlying turn rate $r$ is $1 - e^{-r\Delta t}$; in the limit of short $\Delta t$, this reduces to $r\Delta t$. The probability of not observing a turn is $e^{-r\Delta t}$. Therefore given a model of the turn rate, the probability of observing a particular experimental outcome is given by

$$\log(P(data|model)) = \sum_{turn} \log(r(x)\Delta t) - \sum_{no\ turn} r(x)\Delta t,$$

where $x$ is the filtered signal, $r(x)$ is the turn rate predicted by the model, and $\Delta t$ is the sampling rate. $\sum_{no\ turn}$ is the sum over all points when larvae were in runs and thus capable of initiating turns. We used the MATLAB function fmincon to find the parameters that maximized this log-likelihood.

For *Figures 3C, 4* (cyan line), the separate pathways model rate function was given by

$$r(x) = \bar{r}_O \frac{e^{-\frac{(x_O - \mu_O)^2}{2\sigma_O^2}}}{\sigma_O e^{-\frac{x^2}{2}}} + \bar{r}_L \frac{e^{-\frac{(x_L - \mu_L)^2}{2\sigma_L^2}}}{\sigma_L e^{-\frac{x^2}{2}}},$$

with fit parameters $\bar{r}_O, \sigma_O, \mu_O, \bar{r}_L, \sigma_L, and\ \mu_L$.

For *Figures 3D, 4* (magenta line), the early linear combination model rate function was given by

$$r(x) = \bar{r} \frac{e^{-\frac{(u - \mu)^2}{2\sigma^2}}}{\sigma e^{-\frac{u^2}{2}}}; u = \cos(\theta)x_O + \sin(\theta)x_L,$$

with fit parameters $\bar{r}, \sigma, \mu, and\ \theta$.

## Kullback-Leibler divergence

We estimated the KL divergences in *Figure 3* using the k-nearest neighbors algorithm (*Wang et al., 2009*) as implemented in MATLAB by (*Szabó, 2014*). As a check, for *Figure 3B*, we also analytically computed the KL divergence between Gaussian distributions with the same means and (co-)variances as the sampled turn-triggered and stimulus ensembles (*Pillow and Simoncelli, 2006*), and for *Figure 3C,D*, we numerically calculated the KL divergence between the model predictions and a multivariate Gaussian with the same mean and covariance as the measured turn-triggered density. For the univariate distributions of 3B, we also estimated the divergence using an alternate method based on Szegö's theorem (*Ramırez et al., 2009*) as implemented by (*Szabó, 2014*). These estimates are shown in *Table 2*. None of the conclusions of this paper depend on the method of estimation used.

## Acknowledgements

We thank S Nathasha Egodage and Ruben Contreras for assistance with experiments, and Damon Clark, Katherine Nagel, and Andrew Leifer for valuable comments on the manuscript.

# Additional information

## Funding

| Funder | Grant reference | Author |
| --- | --- | --- |
| New York University | Startup | Ruben Gepner, Mirna Mihovilovic Skanata, Natalie M Bernat, Marc Gershow |

The funder had no role in study design, data collection and interpretation, or the decision to submit the work for publication.

## Author contributions

RG, MMS, Conception and design, Acquisition of data, Analysis and interpretation of data, Drafting or revising the article; NMB, Analysis and interpretation of data, Drafting or revising the article; MK, Acquisition of data, Analysis and interpretation of data, Drafting or revising the article; MG, Conception and design, Analysis and interpretation of data, Drafting or revising the article

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
