## [Decision Letter]

Thank you for sending your work entitled “Identification of computations underlying photo-taxis, odor-taxis, and multi-sensory integration” for consideration at *eLife*. Your article has been favorably evaluated by Eve Marder (Senior editor) and three reviewers, one of whom, Ronald L Calabrese, is a member of our Board of Reviewing Editors.

The Reviewing editor and the other reviewers discussed their comments before we reached this decision, and the Reviewing editor has assembled the following comments to help you prepare a revised submission.

The authors present a very intriguing and thorough behavioral analysis of the navigation of *Drosophila* larvae to visual and chemical stimuli, separately and combined. For this analysis they use light and optogenetically induced fictive olfactory stimuli whose derivatives were independent identically distributed Gaussian random variables: a Brownian random walk. In an automated system, they monitor large numbers of larvae to a variety of stimuli and a stimulus combination and apply a Linear-Nonlinear-Poisson (LNP) model. Their Results are consistent with a model where visual and chemical inputs are combined early rather than processed in parallel to effect orienting turns. Their Discussion suggests that the methods used can be generally applied in understanding how neuronal networks effect decisions. The writing is very clear and crisp and the figures well designed. The modeling is well conceived and based on established procedures and it appears competently and prudently applied. The Materials and methods section and the Figure legends are very clear and helpful. These results have important implications for behavioral analysis of decision making in animals and serve as an entry point for further mechanistic studies in the important model system. Moreover, the application of the techniques here can be a model for applications in other systems.

There are some concerns which the reviewers share that should be addressed:

1) To be sure that the blue light is not evoking responses in Chrimson expressing ORNs the authors should perform a control experiment in genetically blind larvae with a NorpA mutation while expressing Chrimson in Or42b neurons and presenting blue light. The need for this control is expressed well by one of the expert reviews:

“My biggest concern is that I am not entirely convinced that there is no cross talk between the visual stimuli and the fictive odors. This is because the channelrhodopsin Chrimson is still highly sensitive to blue light despite being red-shifted in its optimal excitation wavelength. [29] show that blue light is very effective at eliciting spikes in larval neurons even for very brief and dim light levels (Figure 3A). I disagree that simply because the red light is 300 times greater intensity than the blue light, that any blue light response would be 0.3% of the red light responses. This will depend on the light intensities at which the Chrimson saturates. The percentage could be much higher than reported. Additionally, dim blue light was used in the Chrimson experiments to mask out visual responses to the red light. They reference [29] for this approach. The problem is that [29] were using adult flies, in which blue light is known not to penetrate the cuticle. Thus blue light likely never reached the Chrimson molecules in their central neurons and thus blue-light masking is appropriate. This manuscript uses this approach in peripheral neurons in a transparent larva. Thus, this blue light is more likely to activate Chrimson.

The authors would ideally use physiology to demonstrate clear separation of their visual and fictive stimuli. At minimum a behavioral approach should be attempted. Could the authors not genetically blind larvae with a NorpA mutation while expressing Chrimson in Or42b neurons and presenting blue light? If the blue light still evokes turning, then there is cause for concern.”

2) The Discussion does not do justice to the Results, probably a holdover from this being originally conceived as a short communication. One of the expert reviewers has some good suggestions for amplifying the Discussion:

“The Discussion was too short and failed to put these new results in a broader perspective. What do these results reveal if anything about the strategy used by larvae to avoid blue light and go towards sources of attractive odorants? I.e., what does it mean about what happens when larvae are navigating “natural” environments where odorant and light have spatial structures with characteristic lengths, rather than being subjected to signals with Gaussian white noise derivative? How do these results fit in with previous studies? I wished the authors had discussed further the interesting differences they identify between decisions in response to ‘CO2 signals’ and those to ‘ethyl acetate’ and light. They mention the role of speed change in response to CO2 but do not discuss whether there is speed modulation in response to the other two signals.”

3) There is a more minor concern that the Chrimson responses might show adaptation over the course of the experiments. In the absence of a direct electrophysiological demonstration, is there any evidence the authors can provide that the adaptation is not a factor in for example the poor fit of the LNP model for the first 10s of the step response in Figure 2?

---

## [Author Response]

*1) To be sure that the blue light is not evoking responses in Chrimson expressing ORNs the authors should perform a control experiment in genetically blind larvae with a NorpA mutation while expressing Chrimson in Or42b neurons and presenting blue light. The need for this control is expressed well by one of the expert reviews*.

We have carried out the requested control experiment and now include it as Figure 3—figure supplement 3. We appreciate the reviewers suggesting this experiment, as it vividly demonstrates that we are successfully addressing the visual and olfactory systems separately. Rather than the NorpA mutation, we used GMR-hid to ablate larval photoreceptors. Our results show that *GMR-hid*, *Or42a*>*CsChrimson* larva respond robustly to red light and are unable to perceive blue light.

Figure 3 and Figure 4 also provide direct evidence that dim blue light activates mainly the visual system and that red light activates mainly CsChrimson. *Or42a>CsChrimson* larvae turn in response to an *increase* in blue light and a *decrease* in red light. This is only possible if the red and blue light activate distinct sensory systems.

For instance, consider the right column of Figure 4. Larvae turn in response to a decrease in red light intensity (panel ii), because a decrease in CsChrimson activation of the Or42a receptor neurons is perceived as a decrease in attractive odor concentration. If blue light were mainly activating CsChrimson, we would expect a simultaneous increase in blue light intensity to at least partially offset the effects of decreasing red light. Instead, we see that an increase in blue light coupled to a decrease in red light actually causes more turning than a decrease in red light alone (v), and we see that a decrease in blue light offsets, rather than enhances, the effects of a red light decrease (viii). The larvae’s observed responses are entirely consistent with red light providing a fictive odor cue and blue light providing a visual cue, but impossible to reconcile with the hypothesis that, at the intensities used in our experiments, both red and blue light activate CsChrimson to comparable degrees. We have included a brief discussion of this point in the text.

In particular, the fact that red and blue light have opposite valences to *Or42a>CsChrimson* larvae means that the odor-light integration model we derive cannot result from cross-activation of CsChrimson by blue light.

The expert reviewer also makes several astute technical points which we address below. We are afraid that this discussion might prove somewhat confusing for a more general audience, so we have not included it in our main text.

*“My biggest concern is that I am not entirely convinced that there is no cross talk between the visual stimuli and the fictive odors. This is because the channelrhodopsin Chrimson is still highly sensitive to blue light despite being red-shifted in its optimal excitation wavelength.*
[29]
*show that blue light is very effective at eliciting spikes in larval neurons even for very brief and dim light levels (**Figure 3A**)*.”

The blue light intensity in Klapoetke Figure 3A was 14 mW/cm^2^, while the blue light intensity in our experiments involving CsChrimson was always < 4 μW/cm^2^, a factor of over 3,000 less. Even accounting for the fact that the stimulus of Figure 3A was pulsed, more blue light was delivered in the 1 ms pulse that failed to evoke EJPs than in 3.5 seconds of our constant blue light background.

In adult flies (Figure 3F), Klapoetke et al. found that the threshold peak intensity to achieve a behavioral response to blue light in *Gr64f>CsChrimson* flies was 3.5 mW/cm^2^, a factor of 1000 greater than the light intensities used in our experiments. These experiments were also carried out with pulsed light, but the average intensity during stimulation was 560 μW/cm^2^, 100 times more intense than the blue light intensity we used. In Figure 3F, Klapoetke et al. show that a 600 ms pulse of blue light with mean intensity of 160 μW/cm^2^ (peak intensity 1 mW/cm^2^) failed to evoke behavioral responses in *Gr64f>CsChrimson* flies. This intensity was a factor of 40 greater than the intensity of our blue light mask*.*

*“I disagree that simply because the red light is 300 times greater intensity than the blue light, that any blue light response would be 0.3% of the red light responses. This will depend on the light intensities at which the Chrimson saturates. The percentage could be much higher than reported*.”

It is possible that the highest intensities in our experiments match the saturation limit of CsChrimson, but if the channel were saturating at significantly lower intensity, our reverse correlation experiments simply wouldn’t work, because for most of the time the larva would be experiencing a constant, rather than time-varying, stimulus.

The red light intensity we used in this work is actually fairly modest compared to that used in Klapoetke and in other labs with which we’ve discussed these experiments. In white-noise experiments not presented in this work, we used a significantly stronger red light intensity (peak 3 mW/cm^2^) to activate CsChrimson and found that reverse correlation still produces beautiful kernels and nonlinear rate functions, indicating that saturation is not a problem over the majority of the range of these experiments either.

Looking only at the data presented in the paper, we observe that *Or42a*>*CsChrimson*, *Or42b*>*CsChrimson*, and *Gr21a*>*CsChrimson* larva all show robust responses to step changes in red light between 343 and 572 μW/cm^2^ (Figure 2). Thus the channel is clearly not saturated at 340 μW/cm^2^. Based on this, we can conservatively set an upper limit of the effect of the ∼3 μW/cm^2^ blue light stimulus at 1% of the red light stimulus.

*“Additionally, dim blue light was used in the Chrimson experiments to mask out visual responses to the red light. They reference*
[29]
*for this approach. The problem is that*
[29]
*were using adult flies, in which blue light is known not to penetrate the cuticle. Thus blue light likely never reached the Chrimson molecules in their central neurons and thus blue-light masking is appropriate. This manuscript uses this approach in peripheral neurons in a transparent larva. Thus, this blue light is more likely to activate Chrimson*.”

We cited Klapoetke et al. here to give credit for the approach of suppressing the visual response to red light by using a dim blue stimulus. However, the success of this approach in the adult fly does support its use in the larva. To mask the startle response to 720 nm light in the Proboscis Extension Reflex (PER) assay, Klapoetke et al. used a random blue visual dot stimulus with mean intensity 4 μW/cm^2^, comparable to the intensity of our constant blue mask. These PER experiments were carried out in flies expressing CsChrimson in peripheral sensory neurons, not central brain neurons, using *Gr64f-Gal4* which labels sweet receptors in the labellum and legs (Dahanukar et al., 2007). These neurons are accessible to blue light, as shown by the fact that Klapoetke et al. were also able to evoke PERs using blue light (Figure 4F, 4G) and by recent experiments in which blue light was used to evoke PERs in *Gr64f>ChR2* flies (French et al., 2015).

The uni-modal CsChrimson experiments of our Figure 2 were conducted with constant blue light and varying red light intensities. In these experiments, we can account for the effect of blue light activation of CsChrimson by adding some constant offset to the red light intensity. The red light intensity ranged from 0 to 911 μW/cm^2^ while the blue light intensity was 3.7 μW/cm^2^. If we assume that blue light is equally effective as red light in activating CsChrimson, we can represent the effect of the constant blue light mask on Chrimson activation by adding 3.7 μW to the time varying red light intensity, so that the red intensity instead ranges from 3.7 to 914.7 μW/cm^2^. Because all of our analysis is carried out on the derivative of light intensity, adding this constant offset has no effect on the data presented or the results derived from them.

*“The authors would ideally use physiology to demonstrate clear separation of their visual and fictive stimuli. At minimum a behavioral approach should be attempted. Could the authors not genetically blind larvae with a NorpA mutation while expressing Chrimson in Or42b neurons and presenting blue light? If the blue light still evokes turning, then there is cause for concern*.”

We now present Figure 3—figure supplement 3 which shows turn-triggered averages for genetically blinded (using GMR-hid) larvae expressing CsChrimson in Or42a neurons and for wild-type larvae not expressing CsChrimson. Both groups of larvae were exposed to the same multi-color stimulus used to generate Figure 3. Blind larvae respond only to red light fluctuations and “non-optogenetic” larvae respond only to blue light fluctuations. We thank the reviewer for suggesting these experiments.

*2) The Discussion does not do justice to the Results, probably a holdover from this being originally conceived as a short communication. One of the expert reviewers has some good suggestions for amplifying the Discussion*:

*“The Discussion was too short and failed to put these new results in a broader perspective. What do these results reveal if anything about the strategy used by larvae to avoid blue light and go towards sources of attractive odorants? I.e., what does it mean about what happens when larvae are navigating “natural” environments where odorant and light have spatial structures with characteristic lengths, rather than being subjected to signals with Gaussian white noise derivative? How do these results fit in with previous studies? I wished the authors had discussed further the interesting differences they identify between decisions in response to “CO2 signals” and those to “ethyl acetate” and light. They mention the role of speed change in response to CO2 but do not discuss whether there is speed modulation in response to the other two signals*.”

We present a greatly expanded Discussion section. We describe how our results fit in with previous measurements of larvae in structured odor environments. We spend considerably more time discussing the possible connection between speed modulation in CO_2_ gradients and the differences we discovered in how the larva uses Gr21a activity to make turning decisions. We also use this section to amplify our discussion of the multi-sensory experiments.

*3) There is a more minor concern that the Chrimson responses might show adaptation over the course of the experiments. In the absence of a direct electrophysiological demonstration*, *is there any evidence the authors can provide that the adaptation is not a factor in for example the poor fit of the LNP model for the first 10s of the step response in*
Figure 2?

We now include Figure 2—figure supplement 1, which shows the analysis for the first 10 minutes and second 10 minutes of each 20 minute experiment separately. The turn-triggered averages are remarkably stable across the course of the experiments. The turn-rates decrease slightly at the high end of the animals’ responses for attractive odor stimulation. However, the turn-rate similarly decreases for visual stimulation, suggesting that this habituation might be a property of the animals’ behavior rather than adaptation of the channel. *Gr21a*>*CsChrimson* larvae actually show increased turning in the second 10 minutes of the experiments, suggesting sensitization rather than adaptation. We have added a paragraph to the manuscript discussing this analysis.

We cannot exclude the possibility that larvae might adapt to variance in sensory input. While the mean light levels and mean light derivatives were matched for the white-noise and step-response experiments, the variance of the signals was much greater for the white-noise experiments. Perhaps this explains the failure of the LNP model to fully predict larvae’s step responses.